# Investigating online activity in UK adolescent mental health patients: a feasibility study using a natural language processing approach for electronic health records

Rosemary Sedgwick ![ORCID] ,[1,2] André Bittar,[3] Herkiran Kalsi,[3] Tamara Barack,[3] Johnny Downs ![ORCID] ,[1,2] Rina Dutta ![ORCID] [2,3]

¹Department of Child and Adolescent Psychiatry, Institute of Psychiatry, Psychology and Neuroscience, King's College London, London, UK
²South London and Maudsley NHS Foundation Trust, London, UK
³Department of Psychological Medicine, Institute of Psychiatry, Psychology and Neuroscience, King's College London, London, UK

**Correspondence to**
Dr Rina Dutta;
rina.dutta@kcl.ac.uk

## ABSTRACT

**Objectives** To assess the feasibility of using a natural language processing (NLP) application for extraction of free-text online activity mentions in adolescent mental health patient electronic health records (EHRs).

**Setting** The Clinical Records Interactive Search system allows detailed research based on deidentified EHRs from the South London and Maudsley NHS Foundation Trust, a large south London Mental Health Trust providing secondary and tertiary mental healthcare.

**Participants and methods** We developed a gazetteer of online activity terms and annotation guidelines, from 5480 clinical notes (200 adolescents, aged 11–17 years) receiving specialist mental healthcare. The preprocessing and manual curation steps of this real-world data set allowed development of a rule-based NLP application to automate identification of online activity (internet, social media, online gaming) mentions in EHRs. The context of each mention was also recorded manually as: supportive, detrimental or neutral in a subset of data for additional analysis.

**Results** The NLP application performed with good precision (0.97) and recall (0.94) for identification of online activity mentions. Preliminary analyses found 34% of online activity mentions were considered to have been documented within a supportive context for the young person, 38% detrimental and 28% neutral.

**Conclusion** Our results provide an important example of a rule-based NLP methodology to accurately identify online activity recording in EHRs, enabling researchers to now investigate associations with a range of adolescent mental health outcomes.

## BACKGROUND

Use of the internet, social media and online gaming is now ubiquitous among adolescents. There are general concerns about the potentially harmful impact of screentime on children and young adolescents health, and particularly their mental health.[1] There are also some more established, specific risks online, such as cyberbullying.[2] Internet use is associated with a wide

### STRENGTHS AND LIMITATIONS OF THIS STUDY

⇒ To the authors' knowledge, this paper is the first of its kind to describe the feasibility and development of an natural language processing application for extraction of online activity mentions in electronic health records (EHRs) for use in research.

⇒ Recording of online activity in free-text EHRs will be dependent on both patient report and the detail of documentation by clinicians and, therefore, may not represent the full extent of young people's online use.

⇒ Information extracted using the methods outlined in this paper could provide valuable avenues for further research into the recorded online activity of young adolescent mental health patients and associations with mental health outcomes.

range of adverse outcomes such as self-harm and suicidal behaviour,[3 4] disordered eating and body image issues[5] and symptoms of attention deficit hyperactivity disorder.[6] Problematic video gaming and social media are also associated with several health issues, such as conduct problems and sedentary behaviour.[7] In addition, there is growing evidence, beyond mental health research, for associations between technology and being overweight or obese,[8] with poorer academic performance[9] and exacerbation of educational inequalities.[10] It is, therefore, imperative for mental health services to understand the role of online activity in the populations they serve.

Internet gaming disorder was added to the fifth Diagnostic Statistical Manual[11] and gaming disorder added to the International Classification of Diseases-11.[12] Age, gender, personality characteristics and parental behaviour may all influence adolescents' choice of games[13] and gaming can be done via a number of different devices, both online and offline. Digital platforms are

commonly used by adolescents and a wealth of information may be shared online, providing opportunities for support, information and education. There are now consensus recommendations that asking about online activities should be part of routine clinical assessments.[14–16] The prevalence with which these are noted in mental health assessments completed in Child and Adolescent Mental Health Services (CAMHS), and the context in which they are recorded, has not been studied to date. The existing evidence for the impact of online activity on adolescents is predominantly from cross-sectional survey data and often includes minimal detail about online activities, often with a focus on amount of use, or defined by terms such as problematic internet use.[17] Given the wide range of social media platforms, devices, games and content on the internet, it will be important to gain a more nuanced and real-world understanding of what adolescents are engaging with online. Studying a clinical population of mental health patients will highlight which disorders may predispose adolescents to negative psychological and social impacts of online activity and also what they find supportive. This study provides valuable contextualising information about the recording of online activity in clinical encounters with adolescent mental health patients.

There are validated measures for smartphone, internet and gaming addiction, the most commonly used being the Smartphone Addiction Scale,[18] Internet Addiction Test and the Chen's Internet Addiction Scale,[19] but these are not widely used by clinicians within the UK and there is significant heterogeneity within the research literature. As these structured scales are not commonly used in clinical practice, they will not be uploaded within structured fields on electronic health records (EHRs). However, in the UK, 83% of children aged 12–15 have a smartphone and 69% have at least one social media profile[20] and adolescents with mental disorders spend more time online than those without a mental disorder.[21] Adolescents may not show symptoms suggestive of behavioural addiction, but this does not mean that they are not engaging in activities that may be harmful.

CAMHS in the UK are usually accessed via primary care referral or emergency services in the case of crisis presentations such as self-harm. The National Institute for Health and Care Excellence provides guidelines, and a framework for mental healthcare and assessment, but the EHR platform that this information is documented on varies between NHS trusts. As part of mental health assessment and follow-up, clinicians will often discuss the adolescent's interests and how they spend their time as well as triggers to a recent episode or relapse, such as cyberbullying. The EHRs, therefore, contain unstructured free-text data about online activity of adolescents in contact with CAMHS. Advances in health informatics mean that information extraction tools can be used to automate the extraction of such information.

Natural language processing (NLP) combines computational linguistics with machine learning to allow analysis of unstructured data. This approach has been used across a variety of clinical specialties and health providers to extract information on symptoms, with mental health as one of the most prevalent target populations for study.[22] NLP has already created opportunities to analyse large textual data sets and can now accurately detect mentions of complex phenomena such as suicidality[23–26] and obsessive compulsive symptoms.[27] This study seeks to answer the question of whether an NLP application can derive information on the similarly complex and broad construct of adolescent mental health patient online activity. This will have implication for researchers wishing to undertake large-scale epidemiological research as well as clinicians who could use this personalised data to inform patient care.

## METHODS

### Data source
The Clinical Records Interactive Search (CRIS) system allows detailed research based on EHRs from the South London and Maudsley NHS Foundation Trust, a large south London Mental Health Trust providing secondary and tertiary care to residents of Southwark, Lambeth, Lewisham and Croydon.[28] Care may be provided in mental health settings such as clinics or psychiatric hospitals, or in acute health settings such as emergency departments. In 2014, there were 250 000 patient records.[28] As of September 2019, the EHRs of over 350 000 patients, including over 5.7 million text documents, can be analysed. Clinicians may enter clinical information in a variety of different sections within the EHRs, including: events (unstructured notes), forms (ie, risk assessment) or clinical document attachments such as letters. Events and letters are most commonly used to record clinical information and sometimes the same information may be duplicated across locations.

### Clinical cohort and corpus development
In order to develop an NLP application, it was necessary to generate an adolescent data set within CRIS. Event and attachment documents (n=1 601 422) were derived from 23 455 adolescent patients aged 11–17 in contact with CAMHS between 31 April 2009 and 31 March 2016, as described by Velupillai et al.[29] For the purpose of this paper, n=number of documents, N=number of mentions of online activity. As illustrated in figure 1, from this, a corpus of documents was extracted from a randomly selected group of 200 patients who had a number of EHR documents within the first and third quartiles (document n=5480). This ensured that patients with particularly high or low numbers of records were excluded, as these patients were less likely to be representative of the general clinical population accessing CAMHS, either due to high intensity of contact (such as with prolonged inpatient care) or lack of contact due to non-engagement. Diagnosis was not used as an inclusion or exclusion criterion.

### Mentions of online activity in EHRs
Word searches were a basic but necessary first step to establish prevalence and variability of such terms within free text, especially for such a rapidly evolving and broad construct as online activity. Available literature

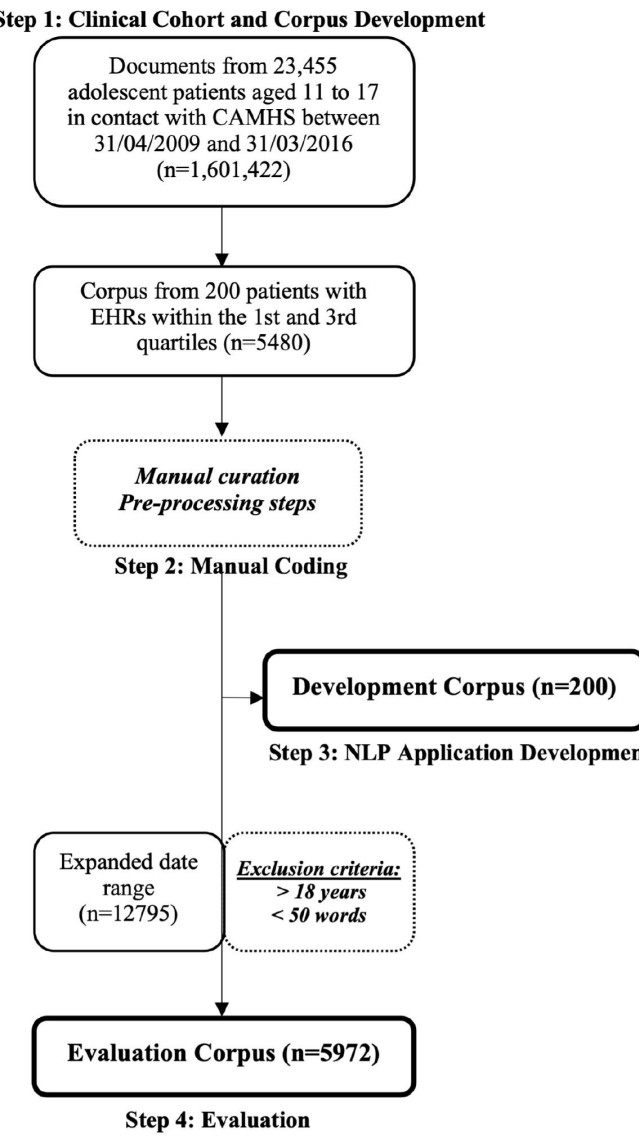

**Figure 1** Method for developing a rule-based NLP application for online activity. n=number of documents. N=number of mentions. CAMHS, Child and Adolescent Mental Health Services; NLP, natural language processing.

| Table 1 | Online activity gazetteer | |
|---|---|---|
| **Social media** | **Internet** | **Online gaming** |
| # | Android | Call of Duty |
| 4chan | Blackberry | Club Penguin |
| askFM | Computer | Computer gam* |
| Bebo | Dark Web | Computer-gam* |
| Blog* | Deep web | Coraline |
| Chatroom* | Googl* | Counter strike |
| cyber-bully* | Internet | Dota 2 |
| cyberbully* | iphone | Dragon age |
| e-communi* | Laptop | Fallout |
| Facebook | Mobile phone | Game Boy |
| Facebook | Online | Game-boy |
| FB | PC | Gaming |
| Flickr | Pinterest | Ghostbusters |
| Forum* | Skype | Grand Theft Auto |
| Hashtag* | Smartphone | HALO |
| Image Sharing | surf* the web | League of legends |
| Instagram | web address | Minecraft |
| Instant messag* | web brows* | Miniclip |
| Linkedin | web surfing | Nintendo |
| lolcow | web-brows* | Online Gam* |
| Myspace | web-surfing | PC gam* |
| Periscope | website* | Playstation |
| Recovery account | you tub* | PS3 |
| Reddit | youtub* | PS4 |
| Snapchat | ipad | Sims |
| Social Media | i-pad | Smite |
| Social Network* | | video game |
| Spam Account | | Wii |
| Tumblr | | World of tanks |
| Tweet* | | World of warcraft |
| Twitter | | Xbox |
| Video sharing | | X-box |
| Vimeo | | Xmen |
| WhatsApp | | X-men |
| Wordpress | | Fortnite |
| | | Pokemon |
| | | Fortnight |
| | | DS |

was searched until there was a saturation of terms. The search included published work, grey literature publications online and policy documents. This was supported by clinical experience from within the research team. This formed the basis of the gazetteer of terms, developed to convey topics that included online devices (ie, computer, iPad), internet terms (eg, websites, specific sites (eg, YouTube), online games (eg, Fortnite), social media terms (eg, forum*) and specific platforms including Facebook, Twitter, and Instagram. The full gazetteer used for the final stages of this research is available in table 1.

### Extracting EHRs for manual curation and pre-processing

The clinical corpus from the inception cohort of 200 patients was used for all subsequent analysis and development. Based on the rationale that a varied lexicon would be used to describe online media use, the gazetteer of key terms was used to identify and filter documents from the corpus with at least one of the search terms, to avoid reading a large volume of unrelated documents. By applying this filter, we identified 217 documents containing at least one of the terms, from 84/200 patients. These were used to gain further contextual insight and identify additional terms relevant to the concepts, including any common misspellings or abbreviations found (ie, Facebook, FB). Documents with one

or more terms from the gazetteer were analysed in detail by two researchers (RS and HK). Many documents were found to be irrelevant 'noise'. Examples were disclaimer messages at the bottom of email contacts or use of the NHS Trust website in letter headers. The term 'email' was found to be generating too much noise for inclusion. Decisions such as this were agreed during regular consensus meetings with the research group.

### Developing manual coding rules

To ensure that future research could be targeted towards more specific exposures, it was necessary to split the search terms to represent three separate classes of mention: internet, social media and online gaming. The class mention might refer to a specific social media platform or game from the gazetteer, or descriptive context, such as 'playing games on the internet'. Further details are found in online supplemental appendix A. The manual curation also identified broad sentiment attributes within clinician documentation: detrimental, supportive or neutral. For example, mention of Facebook in the context of bullying and a subsequent presentation to hospital would be coded as 'SOCIAL MEDIA_DETRIMENTAL'. During the initial scoping exercise, supportive mentions were further split into subcategories to allow for more detailed future analysis and to better capture the context of mentions in the text. This included online activity that adolescents have referred to as supportive, clinician offered supportive advice (eg, recommending online resources) and online activity which supports carers (eg, use of a mental health support forum). Annotation guidelines were developed for the above class and attribute rules to facilitate consistent manual annotation by more than one researcher.

### Manual annotation of online activity and sentiment attributes in EHRs

The preprocessing steps, when applied to the EHRs of the inception cohort, yielded a development corpus of 200 documents from the overall 5480 (derived from 89 of the 200 patients), which formed the data set for the pilot analysis reported below in results. The corpus of 200 documents was divided and annotated for class and attributes by two researchers (RS and HK) using the annotation guidelines. Thirty documents were double annotated and there was an interannotator agreement of kappa coefficient=0.91 for class, 0.68 for attributes and 0.94 for supportive category.[30]

### Development of the online activity NLP application

The preprocessing steps outlined above paved the way for development of the NLP application, designed to automate identification of mentions of online activity use in EHRs. During the manual annotation (human-rater) stage, contextualising online activity raised some challenges. The sentiment attributes were found to be heterogeneous, often lacking detail and more subject to human inter-rater disagreement. Therefore, the algorithm was developed for automation of the class of mention only

(internet, social media or online gaming), based on the manual coding rules applied to the development corpus. Further details and examples are found in the Annotation Guidelines, online supplemental appendix A.

The Online Activity NLP application is a rule-based system based on the spaCy NLP library for Python (V.2.1.3). The application uses four levels of processing, applied sequentially to each document:

1. Text cleaning: removal of 'unwanted' document sections by regular expression replacement.
2. Linguistic preprocessing: sentence and word tokenisation, lemmatisation, and part-of-speech tagging.
3. Lexical annotation: terms in the text are tagged according to the gazetteer (eg, 'computer', 'website' are tagged as INTERNET, 'cyberbully*', 'forum' and 'Instagram' are tagged as SOCIAL_MEDIA) and 'Fortnite' and 'online gaming' are tagged as ONLINE_GAMING.
4. Token sequence annotation: sequences of tokens (ie, words) are annotated and classified (eg, the pattern '(chat|communicat|talk)* online' is tagged as SOCIAL_MEDIA, '(play|playing) fortnite' is tagged as ONLINE_GAMING, etc. This step also removes annotations ('untags') from mentions that were erroneously tagged in the lexical annotation step.

### Patient and public involvement

Development of the gazetteer of online activity terms was supported by face-to-face consultation with adolescent mental health patients through local patient advisory groups up to 2019, including presentation at the Maudsley Biomedical Research Centre YPMHAG.

## RESULTS

The development corpus (n=200) documents extracted through the preprocessing steps (each document containing at least one term from the gazetteer) contained N=243 individual mentions of online activity. In some cases, the same information will be copied into different sections of EHRs but will appear as separate documents. These duplicate mentions, and others that were clearly irrelevant (ie, relating to a typo), were removed (n=115). The remaining 101 documents (64 patients) contained 128 mentions of internet (N=64), social media (N=32), online gaming (N=32). Mean age was 14 (range 11–17 years), from 37 men and 27 women.

### Contextualising mentions of online activity

There were in total 44 supportive mentions (34%), 48 detrimental mentions (38%), 36 neutral mentions (28%). No 'other' mentions recorded in this development corpus. Supportive mentions were subdivided into supportive for the young person (N=25), where a clinician was offering supportive advice (N=17) or where a carer had reported an online activity as helpful (N=2). Each class was also analysed independently to provide pilot data on these different exposures. Internet mentions were 33% detrimental, 48% supportive, 19% neutral.

Social media mentions were predominantly reported by female patients and classed as detrimental (50%), with little supportive benefits (9%). Online gaming was predominantly among male users and showed detrimental (34%), supportive (31%) and neutral (34%) context.

## Evaluation of the online activity NLP application

An evaluation corpus was curated using EHRs from an expanded date range of the inception cohort, from CRIS origination to 2 July 2019. These adolescents were 11–17 at the time of presentation (between 2009 and 2016); therefore, it was anticipated that not all records (n=12 795) would be relevant. As the research group was interested in the CAMHS population specifically, only documents pertaining to adolescents who were still age 18 or younger at the time of documentation were included. Records of less than 50 words were removed due to the lack of relevance. Documents included in the development corpus were also removed (n=200). The remaining evaluation corpus (n=5972) was randomly divided between two researchers (RS and TB) and all relevant mentions of internet/social media/online gaming were manually annotated according to the annotation guidelines (online supplemental appendix A). To establish the human interrater agreement, 200 documents, both with and without annotations were double annotated, yielding a kappa coefficient of 0.94 for annotation of class (internet, social media, online gaming). Sentiment attributes (detrimental, supportive, neutral) were again manually annotated, but as this process was not automated, these were not included in the evaluation. Following a consensus discussion, discrepancies were resolved. These predominantly related to mentions in older documents where limited detail was given, for example, 'playing on the computer'. Adjudicated documents were included to produce a 'gold standard' evaluation corpus containing 535 individual annotations, from 5972 documents. This evaluation revealed a precision of 0.97 and recall of 0.94 and kappa=0.91. Span agreement showed a precision of 0.69 and recall 0.80, with full results available in table 2.

**Table 2** Performance of the online activity NLP application on the evaluation corpus (n=5972)

| Evaluation results | | |
|---|---|---|
| | Span agreement | Class (Internet, Social media, Online gaming) agreement |
| Precision (macro) | 0.69 | 0.97 |
| Recall (macro) | 0.80 | 0.94 |
| F-score (macro) | 0.74 | 0.95 |
| Precision (micro) | N/A | 0.95 |
| Recall (micro) | N/A | 0.95 |
| F-score (micro) | N/A | 0.95 |
| Kappa | N/A | 0.91 |

NLP, natural language processing.

## DISCUSSION

This study provides evidence of the feasibility of using free-text EHR data for the evaluation of online activity in a sample of mental health patients and to the authors' knowledge is the first of its kind to use this methodology. The use of digital interventions in mental health is rapidly growing and there is interest in how these developments should be evaluated in future. In the meantime, it is vital that more evidence-based guidelines reach clinicians to ensure the quality of documentation facilitates research into this important and timely area. This is a move supported by the UK Royal College of Psychiatrists in their 2020 report, which calls for urgent funding of high quality, longitudinal research into the effects of technology on the mental health of young people and the need for technology companies to provide user-generated data for research.[16]

Clinician mentions of online activity will be influenced by the clinician's personal knowledge and experience. The results of our scoping exercises revealed that despite guidance from the British Psychological Association[15] and Royal College of Psychiatrists,[14] the detail of documentation has historically been poor. This is, however, likely to improve, given increasing acknowledgement of the important role of digital technology and mental health for adolescents. As more mental health applications and online resources are recommended by clinicians; familiarity will increase and these discussions will more frequently take place between patients and professionals at CAMHS. Subsequently, recording in free-text EHRs will improve and there may be scope for prospective data collection in the future, prompting clinicians to delineate further detail around online activity use.

The focus of NLP development within CRIS has historically been on symptoms, but detection of behaviours and activities adolescent engage in is also required if we are to better understand the impact on mental health. Automated detection of cyberbullying has been attempted in social media text,[31] and a bullying NLP application has been developed.[32] The NLP application we outline here is important as it will be able to capture emerging online activities and behaviours in the EHRs of adolescents, providing opportunities for much needed longitudinal research into online activity and mental health and well-being, which to date has been lacking. These developments can inform our understanding of the specific risks and benefits of online exposures; inform clinical guidelines and help target future interventions caused by digital exposures and to evaluate interventions delivered through these platforms.

There are no similar studies available for direct comparison and, therefore, the strengths of the NLP application are encouraging. It has shown good precision (0.97) and recall (0.94) in automating detection of mentions of internet, social media and online gaming in our corpus of clinical notes, enabling further research into online activities within a large adolescent mental health population. We divided online activity into broad classes: internet,

social media and online gaming, though it is worth noting that there is increasing overlap to these formats with technological advances. We found that broadly social media was reported in the EHRs as more harmful than online gaming, which may be an important hypothesis generating finding. The concepts of 'supportive', 'detrimental' and 'neutral' have been shown to be promising attributes for automation and incorporation into future iterations of the online activity NLP application. However, limitations to human–coder agreement require further work and a more nuanced taxonomy of terms will be required to accurately reflect the online activity of adolescents.

Detailed analysis about the generalisability of our findings to all mental health patients was outside the scope of this study. The use of unstructured retrospective EHR data has its limitations, in particular, the potential for selection bias. Clinicians may have been more likely to document online activity for certain patient groups who they perceive to be more susceptible to detrimental or supportive impacts, or this may have been influenced by external factors such as publication of professional guidance or individual perception of the importance of these exposures to adolescent mental health. There are limitations to the clinical interpretation possible from our data. The pilot data reported in this paper included any mention of enjoyment as 'supportive' for the young person, unless there was any negating information. Identification and nurturing of enjoyable activities and hobbies can be a useful tool when working with adolescent patients in CAMHS. But, there is also the possibility of these becoming excessive and having a 'detrimental' impact, especially given the increasing concern about gaming disorder.[11 33] Perspectives of the young person and a carer may differ, but this may not be documented by the clinician. It will also be necessary in later iterations of the NLP application to incorporate negating terms and phrases as well as greater sensitivity for which subject (adolescents themselves or third party) the mention of online activity relates to. We found few (n=2) positive mentions of online activity by a parent or carer. Given that young people will be the focus of a clinical encounter, this likely reflects a lack of documentation regarding carer support. This is a limitation to our methodology, given that support and information for carers may increasingly be found online. This and other nuances will become increasingly important on more contemporary data sets, though it is worth noting that it was not considered a major limitation in the historical corpus reported here.

There are limitations to the rule-based approach with such a rapidly evolving field. We endeavoured to have a broad range of search terms in the gazetteer but acknowledge that this is not exhaustive. New games, social media platforms, websites and apps are hard to keep up with, and omission of these titles from the gazetteer may bias studies towards certain online activities. The gazetteer can be added to and amended based on the context of the data and emergence of new popular terms, but this will require a degree of vigilance from users wishing to

apply it to contemporary data sets, or those with other groups, such as adults or disorder-specific cohorts. The application displayed insufficient contextual disambiguation for the following words: computer, internet, mobile phone, online, PC, website. It performed less well-distinguishing class from longer spans of free text, that is, *playing games with friends online* or *playing games on the computer* being incorrectly labelled 'internet' rather than 'online gaming'. Mention of all specific websites described in CRIS would not be feasible, but inclusion of *www.,.co.uk* or other more generic identifiers resulted in too many false positives. Similarly, '*email\**' generated too many false positives during development to be included. These may, therefore, be false negatives that should be considered when using the NLP application, and it is possible that, in some circumstances, precision could be sacrificed for greater recall.

This paper outlines the developments in NLP for use in EHR's within CRIS, but the Online Activity NLP application could also be adapted for other clinical data sets to allow reproduction of results. User-generated data could be another application for this NLP approach and may more accurately capture adolescent online behaviour, unhampered by the recall and reporting bias associated with self-report questionnaires or discussion with a clinician. NLP can already be applied to risk assessment of self-injurious behaviour in user-generated content[34] and the Linguistic Inquiry and Word Count have shown promise in assessing emotional well-being from Facebook posts.[35] Social media data have potential as a rich data source for identification of medical and mental health conditions[36 37] and with further refinement, discussion of risk-associated online activity or 'online harms'[38] could be another avenue for the application of NLP, such as that outlined here. These advancements could eventually lead to earlier detection of at-risk adolescents and targeting of interventions, a field already developing in suicide prevention.[39] There is also scope on a public health level for user-generated content to be useful for communication, monitoring and prediction of disease, which was demonstrated during the COVID-19 pandemic.[40]

There are also potential clinical applications for this work when applied to EHRs. The app could be valuable for characterising online activity patterns for specific patient groups, such as those with eating disorders or Autism Spectrum Disorder, and the impact on later recovery. Since the COVID-19 pandemic, young people's online activity has accelerated, with greater reliance on online means of communication, education and access to mental health support. Our NLP application could provide valuable insight into these trends; providing information on an individual and epidemiological level to guide recommendations. There is also potential for adaptation to more dynamic uses, such as EHR surveillance to track the burden of adverse online experiences through established methods such as Audit and feedback, which can result in important improvements in clinical practice.[41] Information such as this, presented in accessible

ways such as clinician dashboards, could support rapid synthesis of risk factors within an individual or across a service and identify areas of unmet need and potential treatment targets.

## CONCLUSION

We have developed a highly accurate Online Activity NLP application for use in EHRs, which can incorporate keywords as online platforms and services develop over time. This will allow further research using CRIS data to investigate novel risk factor research into a range of adolescent mental health outcomes. It also opens the door for EHR surveillance and clinical monitoring, enabling clinicians to track the burden of adverse online experiences at an individual and service level. Beyond its proven utilisation in EHRs, tools such as this also have the potential to be adapted to other clinical or non-clinical data sets, which could enhance our understanding of these new phenomena and the impact on adolescent health and well-being.

**Acknowledgements** The authors acknowledge infrastructure support from the National Institute for Health Research (NIHR). The views expressed are those of the authors and not necessarily those of the NHS, the NIHR or the Department of Health and Social Care.

**Contributors** RS developed the concept, and led on the study design, data collection, writing of results and final draft of the manuscript. AB developed the NLP application and ran the evaluations. Manual coding rules were written by RS with input from the other authors, including JD and RD. Reviewing of EHRs for the manual curation stage was performed by RS and HK. Reviewing the EHRs for the Evaluation Corpus was performed by RS and TB. All authors were involved in the writing and review of the manuscript. All authors declare no competing conflict of interest. RD acts as guarantor for this study.

**Funding** RS was funded by the National Institute for Health and Social Care Research (NIHR) Academic Clinical Fellowship and a NIHR Biomedical Research Centre at South London and Maudsley NHS Foundation Trust Preparatory Fellowship. JD is supported by NIHR Clinician Science Fellowship award (CS-2018-18-ST2-014) and has received support from a Medical Research Council (MRC) Clinical Research Training Fellowship (MR/L017105/1) and Psychiatry Research Trust Peggy Pollak Research Fellowship in Developmental Psychiatry. RD is funded by an MRC award (grant code: MR/S020365/1); a Clinician Scientist Fellowship from the Health Foundation in partnership with the Academy of Medical Sciences and her work is supported by the NIHR Biomedical Research Centre at South London and Maudsley NHS Foundation Trust and King's College London.

**Disclaimer** This paper represents independent research part-funded by the NIHR Maudsley Biomedical Research Centre at South London and Maudsley NHS Foundation Trust and King's College London. The views expressed are those of the author(s) and not necessarily those of the NIHR or the Department of Health and Social Care. For the purposes of open access, the author has applied a Creative Commons Attribution (CC BY) licence to any Accepted Author Manuscript version arising from this submission.

**Competing interests** RD declares previous research funding received from Janssen. The remaining authors declare that the research was conducted in the absence of any commercial or financial relationships that could be construed as a potential conflict of interest.

**Patient and public involvement** Patients and/or the public were involved in the design, or conduct, or reporting, or dissemination plans of this research. Refer to the Methods section for further details.

**Patient consent for publication** Not applicable.

**Ethics approval** The use of CRIS data for research was approved by the Oxfordshire Research Ethics Committee C (reference 08/H0606/71 + 5). CRIS data

is used by researchers in a de-identified and data-secure format and patients have the choice to opt-out of their data being used. CRIS approval for this project has been granted by the CRIS oversight committee (Project reference: 18-102) and all data for use in this research has been accessed in accordance with CRIS Governance procedures.

**Provenance and peer review** Not commissioned; externally peer reviewed.

**Data availability statement** Data may be obtained from a third party and are not publicly available. The data accessed by CRIS remain within an NHS firewall and governance is provided by a patient-led oversight committee. Access to data is restricted to honorary or substantive employees of the South London and Maudsley NHS Foundation Trust and is governed by a local oversight committee who review and approve applications to extract and analyse data for research. Subject to these conditions, data access is encouraged and those interested should contact the CRIS academic lead.

**ORCID iDs**
Rosemary Sedgwick http://orcid.org/0000-0002-6344-9353
Johnny Downs http://orcid.org/0000-0002-8061-295X
Rina Dutta http://orcid.org/0000-0002-5614-8659

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
