## [Reviewer comments · BMJ Open]

ARTICLE DETAILS

TITLE (PROVISIONAL)	Investigating online activity in UK adolescent mental health patients: a feasibility study using a Natural Language Processing approach for Electronic Health Records
AUTHORS	Sedgwick, Rosemary; Bittar, André; Kalsi, Herkiran; Barack, Tamara; Downs, Johnny; Dutta, Rina

VERSION 1 – REVIEW

REVIEWER	Gunasekeran, Dinesh Visva Singapore General Hospital
REVIEW RETURNED	26-May-2022

GENERAL COMMENTS	General comments - This paper describes and evaluates an attempt to develop a method for evaluating this increasingly important element of online activity modulating mental health, using scalable NLP tools and retrospective clinical entries in EHRs. Major comments 1. The manuscript is informative for readers and attempts to develop a method for longitudinal impact of online activity on CYP based on retrospective EHR data. In the opening paragraph of the Introduction the authors outline various negative examples of online activity. However, children may also have constructive use of online platforms such as Youtube for learning free courses. Further details about pre-processing steps may help readers interpret data and iterate on methodology in future studies. For example, in case positive examples may be under represented by virtue of the sentiment annotation approach used (results n=2 instances of helpful online activity among supportive mentions group). 2. The use of unstructured retrospective entry information about online activity has several limitations. For instance, there is potential selection bias if clinicians had only documented online activity for specific profiles of patients, that would then influence the classification of the NLP algorithm. There are limits to the accuracy of retrospective EHR data that should also be acknowledged in discussion, particularly unstructured and subjective data such as online activity. If there is a lack of framework for documentation and interpretation with clinical relevance (depending when the dates of the data were collected versus the propagation of the academic consensus cited by the authors – reference 14-16 -into practice), then there may not be any established practice for the documentation to guarantee accuracy, and even the decision whether to document at all
--

	contributes subjectivity. Perhaps these limitations are worthy of discussion, and future investigations could utilise prospective data entry that is prompted with instructions to delineate in detail the nature and duration of online activity to draw clinically relevant findings. They could also triangulate data from real-world online activity with patients' consent. This could be discussed as potential future directions along with relevant existing literature. For example, a recent review regarding public health applications of social media outlined the potential use of machine learning to analyse sentiment and detect symptom trends based on real-world user generated content (UGC) – PMID: 35129456; https://doi.org/10.2196/33680 3. In the opening line of Discussion section, “This study provides evidence of the feasibility of using free-text EHR data for the evaluation of online activity in mental health patients and to the authors knowledge is the first of its kind to use this methodology” may need to be clarified to reflect the pilot nature of the study design, and the true range of patients included. With the exclusion of record sampling for patients with low engagement and patients with high intensity of contact as described in methods, results may not be generalisable to all mental health patients. Perhaps this would be reflective of a subset of patients with mild-moderate disease or those managed in outpatient settings. Similarly, for Conclusion, “We have developed a highly accurate Online Activity NLP application for use in EHRs, which can incorporate keywords as online platforms and services develop over time.”. Minor comments 1. Try to remove acronyms if not essential e.g. “CYP” was used interchangeably with “adolescent” Conclusion: The reviewer’s recommendation is to accept with minor revisions given the timely and important scope of this manuscript
--	--

REVIEWER	Srivastava, Gautam Brandon University
REVIEW RETURNED	26-Jul-2022

GENERAL COMMENTS	In this paper, the authors assess the feasibility of using a Natural Language Processing (NLP) application for extraction of free text online activity mentions in adolescent mental health patient Electronic Health Records (EHRs). This is a well studied domain. The authors have failed to establish a clear research gap that they are trying to fill. Moreover, the authors have missed many key references and related work in the area that could help to establish a research gap they could fill. I suggest a re-rewrite focusing on a distinct issue in NLP EHR, not as broad as presented. And specific experimentation to accompany such an issue.
---

REVIEWER	Dowell, Anthony Wellington School of Medicine and Health Sciences, General Practice
REVIEW RETURNED	18-Oct-2022

GENERAL COMMENTS	This paper brings together two important research themes, the impact of online activity on adolescent mental health, and the use of Natural Language Processing algorithms. The paper thus has potential relevance to a wide range of health professionals. Background. The background begins by an appropriate review of current literature relating to internet and social media use among children and young adolescents (CYP) with particular mention of the potential harm and negative aspects. The authors build a case for their study by identifying a lack of current detailed information about different types of online activities and the fact that clinicians are unlikely to systematically use available validated measures to record information in this area. As elsewhere in clinical practice, much information is held in unstructured and uncoded narratives within the electronic health record, and the authors highlight the potential to extract this information with automated machine learning methods such as NLP.  • While highlighting the application of NLP in mental health settings, for a general readership it might be worth making reference to the application of NLP from EHR in other areas of clinical practice, and the potential it offers . Koleck TA, Dreisbach C, Bourne PE, Bakken S. Natural language processing of symptoms documented in free-text narratives of electronic health records: a systematic review. Journal of the American Medical Informatics Association. 2019 Apr;26(4):364-79. Methods There is clear detail provided of the different stages of the methodology including the data sources and the way that the data set is prepared for NLP algorithm development and training. Appropriate emphasis is given on the ethics and data security of the data sources, given the mental health details within the EHR. There is sufficient detail provided for the general reader to understand the various processes required in the production of the search term gazetteer and its use in the development of the NLP application. For general readership it would be helpful to clarify that the algorithm was able to distinguish between positive and negative mentions of the search terms.  • Given the increasing importance of service user co-design principles in mental health research additional information about the consultation process with adolescent mental health patients (patient and public involvement), would be helpful. Results There is a logical flow of results beginning with a description of findings from the development corpus, the balance of supportive, detrimental and neutral mentions and the demographic distribution of the comments. This is then linked to the results of the NLP application and the comparison with 'gold standard' adjudication, and a conclusion indicating that the algorithm could have feasible clinical and research application. The results also highlight the relatively low level of recording of these issues in the EHR data set. Discussion The discussion provides an appropriate commentary on both the feasibility of using free text EHR data to explore the extent of online activity by a cohort of CYP mental health service users and
---

	also the extent to which historically the topic has been recorded by clinicians.  • As noted above it would be helpful to make reference to the use of NLP applications in other areas of clinical practice. This is important not just for research and evaluation, but also the potential for routine surveillance of particular issues. The authors mention automated detection of cyberbullying , but there are potential other surveillance opportunities. Strengths and limitations are appropriately identified, with the challenge of rapidly evolving gazetteer terms a particular challenge.
--	---

REVIEWER	Levis, Maxwell White River Junction VA Medical Center
REVIEW RETURNED	20-Oct-2022

GENERAL COMMENTS	Thanks for the opportunity to review this elegant manuscript about the development on an NLP methodology for analyzing internet behavior in youth mental health populations. Some feedback is listed as follows: Introduction and background section point towards a focus on youth internet usage and associated risks and concerns. It would help to make clearer that the focus is on a clinical subset of this population from earlier in the section. It would be helpful to provide some additional information about EHR utilization in the UK and more broadly about youth mental health services. Some additional background on NLP and how method works would aid argumentation. Methods: More information about derivation of broad sentiment attributes within clinician documentation would be helpful. Clarification about the development of the development corpus would be useful. Is this all of the documents of the 89 that had indication of the original terms? Supportive category appears to be one of the sentiment attributes, but is listed with distinct kappa statistics. Could you clarify this derivation? Were any notes/ patients removed from initial set due to noise/ false identification? More information in text about “class of mention” would be helpful, as I imagine there would be high amounts of class overlap. More information about the Patient and public involvement component is necessary. I recognize that this subsample was selected without diagnostic consideration, but would be helpful to know more about this population. Small issues: Could you clarify this number: “In 2014 there were 250,000 million”? Why is events in quotes while other formats are not? Similarly for ‘neutral’? Would help to clarify if the term attached vs attachment documents? Removing other close word variances would aid understandability. What are letters in this situation? While it is clear from figure, it would help to explain quartiles in text. Results: Please clarify the statement “Duplicate and irrelevant mentions were removed”. How is “ethnicity representative of the local population” verified? I am a little confused regarding inclusion of sentiment attributes. It seemed like this section was dropped, but
--

	results are presented. However, they are not presented in the "Evaluation of the Online Activity NLP application" section. Small issues: Social media mentions were predominantly female? Discussion: This section is clear but would help to address why sentiment attribute had limited results. Also, would help to address limitations of EHR formats and clinician bias. How do coming changes in structured and unstructured formats impact relevance? How would model be adapted for non-clinical sample? Can you offer some additional support from relevant other NLP studies in terms of sample size? Small issues: "Increasing overlap? It is an "app" or "application"?"
--	--

REVIEWER	Stubbe, Maria University of Otago, Primary Health Care and General Practice
REVIEW RETURNED	24-Oct-2022

GENERAL COMMENTS	A clearly written article reporting on the successful development and piloting of an NLP tool for examining free text in EHRs relating to adolescents' use of the internet and social media. I would have liked to see a little more commentary in the discussion on how the declared limitations relating to accurate categorisation of longer text strings might be addressed in future iterations of this tool, and how often new terms will be added to the lexicon, and how these will be validated.
---

REVIEWER	Magrangeas, Thibault King's College London Institute of Psychiatry Psychology and Neuroscience, Department of Psychological Medicine
REVIEW RETURNED	01-Nov-2022

GENERAL COMMENTS	Thank you for this very interesting study and your clear presentation of methods and outcomes. This provides valuable insights, I simply have a few points I was hoping you may be able to elaborate on: - You mention in your abstract the result of your preliminary analyses. From what I understand, these results are drawn from the 200 document testing sample but there was significant discrepancy in the attribute inter-rater agreement and this analysis of positive, neutral or negative effects was not carried out on the wider sample. You mention this in your discussion but if my understanding is correct (please do correct me if I'm mistaken!), could you include this in the abstract result section? 1 - The current phrasing of your results presentation in the abstract seems to imply the preliminary results discussed in my previous point are the main outcome of this study. Could you rephrase to emphasise more the success of your NLP application focusing on identifying online activity, which appears to be the main outcome? 2 - In lines 22-24 of page 12, you mention that "Following a consensus discussion, discrepancies were resolved", would it be possible to elaborate on any specific issues that led to discrepancies if such an issue was identified, or were these disagreements broad with no consistent theme? 3 - The patient sample you chose is very representative of the teenage population, covering a broad group going from ages 11 to
---

	17. Were there any differences across ages or gender (eg early teenagers vs late) in activities and engagement? You mention differences in gender with girls engaging with social media more and boys with online gaming, but did this change across age groups and if so would it be worth presenting in a short table/diagram? 4 - Similarly, considering the broadness of these categories, is it possible to elaborate on whether certain specific social media or games had more or less of a positive/negative effect on people, or would this require a second follow-up study (again, your manuscript constitutes very interesting results on a new topic and I appreciate a wider scope may require further studies) 5 - The gazetteer you used covers a good and broad selection of terms. Was there a specific reason why some terms eg Xmen were included and not others that I imagine would be prevalent such as Marvel? Similarly, were some terms used found to not be as relevant as may have been expected from existing documentations you used to develop your gazetteer, eg blackberry vs iPad, in which case these documents and guidelines may require an update? 6 - I find the figure of 250,000 million documents (amounting to 250 billion) in line 35 page 5 hard to believe - could you confirm if this is accurate or should instead be 250 million documents? 7 - Grammar: please could you check appropriate use of apostrophes eg "to the authors knowledge" (line 60 page 2) should be "to the authors' knowledge" (repeated several times in the manuscript) Thank you again for this manuscript, I look forward to reading any updates you may have.
--	--

VERSION 1 – AUTHOR RESPONSE

Reviewer: 1 Dr. Dinesh Visva Gunasekeran, Singapore General Hospital	
Comments to the Author	Author Response
General comments - This paper describes and evaluates an attempt to develop a method for evaluating this increasingly important element of online activity modulating mental health, using scalable NLP tools and retrospective clinical entries in EHRs.	N/A
1.a) The manuscript is informative for readers and attempts to develop a method for longitudinal impact of online activity on CYP based on retrospective EHR data. In the opening paragraph of the Introduction the authors outline various negative examples of online activity. However, children may also have constructive use of online platforms such as Youtube for learning free courses.	a) We have provided further emphasis to the possibility for constructive online activity in the introduction: "Digital platforms are commonly used by adolescents and a wealth of information may be shared online, providing opportunities for support, information and education."

b) Further details about pre-processing steps may help readers interpret data and iterate on methodology in future studies. For example, in case positive examples may be under represented by virtue of the sentiment annotation approach used (results n=2 instances of helpful online activity among supportive mentions group).	b) We have included some further detail about this in the discussion: “We found few (n=2) positive mentions of online activity by a parent or carer. Given that young people will be the focus of a clinical encounter, this likely reflects a lack of documentation regarding carer support. This is a limitation to our methodology, given that support and information for carers may increasingly be found online.”
2. The use of unstructured retrospective entry information about online activity has several limitations. For instance, there is potential selection bias if clinicians had only documented online activity for specific profiles of patients, that would then influence the classification of the NLP algorithm. There are limits to the accuracy of retrospective EHR data that should also be acknowledged in discussion, particularly unstructured and subjective data such as online activity. If there is a lack of framework for documentation and interpretation with clinical relevance (depending when the dates of the data were collected versus the propagation of the academic consensus cited by the authors – reference 14-16 -into practice), then there may not be any established practice for the documentation to guarantee accuracy, and even the decision whether to document at all contributes subjectivity. Perhaps these limitations are worthy of discussion, and future investigations could utilise prospective data entry that is prompted with instructions to delineate in detail the nature and duration of online activity to draw clinically relevant findings. They could also triangulate data from real-world online activity with patients’ consent. This could be discussed as potential future directions along with relevant existing literature. For example, a recent review regarding public health applications of social media outlined the potential use of machine learning to analyse sentiment and detect symptom trends based on real-world user generated content (UGC) – PMID: 35129456; https://doi.org/10.2196/33680	Thank you for these comments which we have tried to address with the following inclusions to the Discussion section: “Subsequently, recording in free-text EHRs will improve and there may be scope for prospective data collection in the future, prompting clinicians to delineate further detail around online activity use.” “The use of unstructured retrospective EHR data has its limitations, in particular the potential for selection bias. Clinicians may have been more likely to document online activity for certain patient groups who they perceive to be more susceptible to detrimental or supportive impacts, or this may have been influenced by external factors such as publication of professional guidance or individual perception of the importance of these exposures to adolescent mental health.” The potential for use of real-world online activity data has been discussed in some detail towards the end of the discussion. This interesting paper has been referenced though: “There is also scope on a public health level for user generated content to be useful for communication, monitoring and prediction of disease, which was demonstrated during the COVID-19 pandemic” (reference 40).
3. In the opening line of Discussion section, “This study provides evidence of the feasibility of using free-text EHR data for the evaluation of online activity in mental health patients and to the authors knowledge is the first of its kind to use this methodology” may need to be clarified to reflect	The following sections in the discussion have been added or amended: “This pilot study provides evidence of the feasibility of using free-text EHR data for the evaluation of online activity in sample of mental health patients and to the authors’

the pilot nature of the study design, and the true range of patients included. With the exclusion of record sampling for patients with low engagement and patients with high intensity of contact as described in methods, results may not be generalisable to all mental health patients. Perhaps this would be reflective of a subset of patients with mild-moderate disease or those managed in outpatient settings. Similarly, for Conclusion, “We have developed a highly accurate Online Activity NLP application for use in EHRs, which can incorporate keywords as online platforms and services develop over time.”	knowledge is the first of its kind to use this methodology.” “Detailed analysis about the generalisability of our findings to all mental health patients is outside the scope of this study.”
Minor comments 1. Try to remove acronyms if not essential e.g. “CYP” was used interchangeably with “adolescent”	CYP has been removed and changed to adolescent consistently throughout.
Conclusion: The reviewer’s recommendation is to accept with minor revisions given the timely and important scope of this manuscript	N/A

Reviewer 2: Dr. Gautam Srivastava, Brandon University	
Comments to the Author:	Author Response:
In this paper, the authors assess the feasibility of using a Natural Language Processing (NLP) application for extraction of free text online activity mentions in adolescent mental health patient Electronic Health Records (EHRs). This is a well-studied domain. The authors have failed to establish a clear research gap that they are trying to fill. Moreover, the authors have missed many key references and related work in the area that could help to establish a research gap they could fill. I suggest a re-rewrite focusing on a distinct issue in NLP EHR, not as broad as presented. And specific experimentation to accompany such an issue.	We thank Dr Srivastava for taking the time to review our paper and hope that in addressing the other five reviewers specific comments and recommendations we have provided sufficient evidence of the value of this paper.

Reviewer: 3 Prof. Anthony Dowell, Wellington School of Medicine and Health Sciences	
Comments to the Author:	Author Response:
This paper brings together two important research themes, the impact of online activity on adolescent mental health, and the use of Natural Language Processing algorithms. The paper thus has potential relevance to a wide range of health professionals.	N/A
Background. The background begins by an appropriate review of	

current literature relating to internet and social media use among children and young adolescents (CYP) with particular mention of the potential harm and negative aspects. The authors build a case for their study by identifying a lack of current detailed information about different types of online activities and the fact that clinicians are unlikely to systematically use available validated measures to record information in this area. As elsewhere in clinical practice, much information is held in unstructured and uncoded narratives within the electronic health record, and the authors highlight the potential to extract this information with automated machine learning methods such as NLP. While highlighting the application of NLP in mental health settings, for a general readership it might be worth making reference to the application of NLP from EHR in other areas of clinical practice, and the potential it offers . Koleck TA, Dreisbach C, Bourne PE, Bakken S. Natural language processing of symptoms documented in free-text narratives of electronic health records: a systematic review. Journal of the American Medical Informatics Association. 2019 Apr;26(4):364-79.	Thank you for this helpful reference which we have now included: “This approach has been used across a variety of clinical specialties and health providers to extract information on symptoms, with mental health as one of the most prevalent target populations for study” (reference 22)
Methods There is clear detail provided of the different stages of the methodology including the data sources and the way that the data set is prepared for NLP algorithm development and training. Appropriate emphasis is given on the ethics and data security of the data sources, given the mental health details within the EHR. There is sufficient detail provided for the general reader to understand the various processes required in the production of the search term gazetteer and its use in the development of the NLP application. For general readership it would be helpful to clarify that the algorithm was able to distinguish between positive and negative mentions of the search terms. Given the increasing importance of service user co-design principles in mental health research additional information about the consultation process with adolescent mental health patients (patient and public involvement), would be helpful.	For reasons discussed in more detail in the discussion sentiment attributes were not evaluated using the NLP tool, we did however think that the preliminary data would be of interest to readers, as this will clearly be an avenue for further research. We have included in the Evaluation section of results: “Sentiment attributes were again manually annotated, but these were not included in the evaluation.” Further details about the groups consulted has been included: “This was supported by clinical experience from within the research team and consultation with adolescents through

	face-to-face interactions at local patient advisory groups, including Maudsley BRC Young People’s Mental Health Advisory Group (YPMHAG).”
Results There is a logical flow of results beginning with a description of findings from the development corpus, the balance of supportive, detrimental and neutral mentions and the demographic distribution of the comments. This is then linked to the results of the NLP application and the comparison with ‘gold standard’ adjudication, and a conclusion indicating that the algorithm could have feasible clinical and research application. The results also highlight the relatively low level of recording of these issues in the EHR data set.	N/A
Discussion The discussion provides an appropriate commentary on both the feasibility of using free text EHR data to explore the extent of online activity by a cohort of CYP mental health service users and also the extent to which historically the topic has been recorded by clinicians. As noted above it would be helpful to make reference to the use of NLP applications in other areas of clinical practice. This is important not just for research and evaluation, but also the potential for routine surveillance of particular issues. The authors mention automated detection of cyberbullying , but there are potential other surveillance opportunities. Strengths and limitations are appropriately identified, with the challenge of rapidly evolving gazetteer terms a particular challenge.	We have included the suggested reference above.

Reviewer: 4 Dr. Maxwell Levis, White River Junction VA Medical Center	
Comments to the Author:	Author Response:
Thanks for the opportunity to review this elegant manuscript about the development on an NLP methodology for analyzing internet behavior in youth mental health populations. Some feedback is listed as follows:	Many thanks to Dr Levis for the helpful comments.
Introduction and background section point towards a focus on youth internet usage and associated risks and concerns. It would help to make clearer that the focus is on a clinical subset of this population from earlier in the section.	We have included mention of mental health population earlier in this section: “It is therefore imperative for mental health services to understand the role of online activity in the populations they serve.”

It would be helpful to provide some additional information about EHR utilization in the UK and more broadly about youth mental health services. Some additional background on NLP and how method works would aid argumentation.	To provided further context on access to services: “CAMHS in the UK are usually accessed via primary care referral, or emergency services in the case of crisis presentations such as self-harm. The National Institute for Health and Care Excellence provides guidelines, and a framework for mental health care and assessment, but the EHR platform that this information is documented on varies between NHS trusts..” Re. NLP, hopefully this provides some clarification for the background section (as more detail are outlined later in the methods):“Advances in health informatics mean that information extraction tools can be used to automate the extraction of such information. Natural Language Processing (NLP) combines computational linguistics with machine learning to allow analysis of this previously unstructured data.”
Methods: More information about derivation of broad sentiment attributes within clinician documentation would be helpful. Clarification about the development of the development corpus would be useful. Is this all of the documents of the 89 that had indication of the original terms? Supportive category appears to be one of the sentiment attributes, but is listed with distinct kappa statistics. Could you clarify this derivation? Were any notes/ patients removed from initial set due to noise/ false identification? More information in text about “class of mention” would be helpful, as I imagine there would be high amounts of class overlap.	Thank you to Dr Levis for these detailed points on the methods. I have tried to address them here, but am happy to try and provide further clarification if required. Please refer to appendix A for further details about how we annotated sentiment attributes. This was supported by consensus meetings within the research group. The development corpus was a corpus of documents with at least one mention of online activity, as described in the annotation guidelines (Appendix A). In figure 1 we have attempted to illustrate this to aid clarity on how the Development Corpus and the Evaluation Corpus were developed. Unlike DETRIMENTAL, NEUTRAL or OTHER, supportive category was sub-divided (please see Appendix A for further details) which resulted in a distinct kappa value. The Development Corpus only contained relevant documents, selected through the manual curation and pre-processing steps outlined.

More information about the Patient and public involvement component is necessary. I recognize that this subsample was selected without diagnostic consideration, but would be helpful to know more about this population.	Examples of this were made available in Appendix A and I have now added some further context into the main text: “The class mention might refer to a specific social media platform or game from the gazetteer, or descriptive context, such as “Playing games on the internet”. Further details can be found in Appendix A” Further details about the groups consulted has been included: “This was supported by clinical experience from within the research team and consultation with adolescents through face-to-face interactions at local patient advisory groups, including Maudsley BRC Young People’s Mental Health Advisory Group (YPMHAG).” Unfortunately this level of analysis is outside the scope of this paper. However, our plan is that this methodology could be further developed for future research investigating online activity in specific patient groups in more detail.
Small issues: Could you clarify this number: “In 2014 there were 250,000 million”? Why is events in quotes while other formats are not? Similarly for ‘neutral’? Would help to clarify if the term attached vs attachment documents? Removing other close word variances would aid understandability. What are letters in this situation? While it is clear from figure, it would help to explain quartiles in text.	These typos have been corrected. This sentence has been added to the methods section, which corresponds with the information in figure 1: “For the purpose of this paper n=number of documents, N=number of mentions of online activity.”
Results: Please clarify the statement “Duplicate and irrelevant mentions were removed”. How is “ethnicity representative of the local population” verified?	Sentence expanded to clarify the duplicate and irrelevant mentions: “In some cases the same information will be copied into different sections of EHR’s but will appear as separate documents. These duplicate mentions, and others that were clearly irrelevant (i.e. relating to a typo) were removed (n=115).” To avoid confusion given that detailed analysis of demographic and diagnostic characteristics were not in the scope of this paper, I have removed this statement and we hope that the methodology gives sufficient information to contextualise the population. In

I am a little confused regarding inclusion of sentiment attributes. It seemed like this section was dropped, but results are presented. However, they are not presented in the "Evaluation of the Online Activity NLP application" section. Small issues: Social media mentions were predominantly female?	the discussion I have also been clear of this limitation: "Detailed analysis about the generalisability of our findings to all mental health patients was outside the scope of this study." For reasons discussed in more detail in the discussion sentiment attributes were not evaluated using the NLP tool, we did however think that the preliminary data would be of interest to readers, as this will clearly be an avenue for further research. We have included in the Evaluation section of results clarity: "Sentiment attributes were again manually annotated, but these were not included in the evaluation." - sentence amended: "Social media mentions were predominantly reported by female patients.."
Discussion: This section is clear but would help to address why sentiment attribute had limited results. Also, would help to address limitations of EHR formats and clinician bias. How do coming changes in structured and unstructured formats impact relevance? How would model be adapted for non-clinical sample? Can you offer some additional support from relevant other NLP studies in terms of sample size?	In the text we have explained that at this stage we were not able to automate sentiment attributes. "The sentiment attributes were found to be heterogeneous, often lacking detail and more subject to human inter-rater disagreement." Further work on sentiment attributes and feasibility of using the application in non-clinical samples will be areas of future study. In response to another reviewers comment we have now included the following in the Discussion to further acknowledge these limitations: "Detailed analysis about the generalisability of our findings to all mental health patients was outside the scope of this study. The use of unstructured retrospective EHR data has its limitations, in particular the potential for selection bias. Clinicians may have been more likely to document online activity for certain patient groups who they perceive to be more susceptible to detrimental or supportive impacts, or this may have been influenced by external factors such as publication of professional guidance or individual perception of the importance of these exposures to adolescent mental health."
Small issues: "Increasing overlap?"	Sentence expanded: "We divided online activity into broad classes: internet, social media, and online gaming, though it is worth

It is an “app” or “application”?	noting that there is increasing overlap to these formats with technological advances.” Application
--	--

Reviewer: 5 Dr. Maria Stubbe, University of Otago	
Comments to the Author:	Author Response:
Comments to the Author: A clearly written article reporting on the successful development and piloting of an NLP tool for examining free text in EHRs relating to adolescents’ use of the internet and social media. I would have liked to see a little more commentary in the discussion on how the declared limitations relating to accurate categorisation of longer text strings might be addressed in future iterations of this tool, and how often new terms will be added to the lexicon , and how these will be validated.	We would like to thank Dr Stubbe for her feedback on our paper. Given the exploratory nature and scope of this study we have not included further specific details about future directions at this stage, as this will be the subject of future publications.

Reviewer: 6: Dr. Thibault Magrangeas, King’s College London Institute of Psychiatry Psychology and Neuroscience	
Comments to the Author:	Author Response:
Thank you for this very interesting study and your clear presentation of methods and outcomes. This provides valuable insights, I simply have a few points I was hoping you may be able to elaborate on:	
You mention in your abstract the result of your preliminary analyses. From what I understand, these results are drawn from the 200 document testing sample but there was significant discrepancy in the attribute inter-rater agreement and this analysis of positive, neutral or negative effects was not carried out on the wider sample. You mention this in your discussion but if my understanding is correct (please do correct me if I’m mistaken!), could you include this in the abstract result section? 1 - The current phrasing of your results presentation in the abstract seems to imply the preliminary results discussed in my previous point are the main outcome of this study. Could you rephrase to emphasise more the success of your	We have included mention of the sentiment analysis being “in a subset of data” in the methods and that the precision and recall was “for identification of online activity mention” which hopefully provides sufficient clarification.

NLP application focusing on identifying online activity, which appears to be the main outcome?	
2 - In lines 22-24 of page 12, you mention that "Following a consensus discussion, discrepancies were resolved", would it be possible to elaborate on any specific issues that led to discrepancies if such an issue was identified, or were these disagreements broad with no consistent theme?	Further detail added" "Following a consensus discussion, discrepancies were resolved. These predominantly related to mentions in older documents where limited detail was given, for example "playing on the computer"."
3 - The patient sample you chose is very representative of the teenage population, covering a broad group going from ages 11 to 17. Were there any differences across ages or gender (eg early teenagers vs late) in activities and engagement? You mention differences in gender with girls engaging with social media more and boys with online gaming, but did this change across age groups and if so would it be worth presenting in a short table/diagram?	We feel that this is outside the scope of this pilot paper given its exploratory nature, but analysis such as this would be the aim of future work.
4 - Similarly, considering the broadness of these categories, is it possible to elaborate on whether certain specific social media or games had more or less of a positive/negative effect on people, or would this require a second follow-up study (again, your manuscript constitutes very interesting results on a new topic and I appreciate a wider scope may require further studies)	As above.
5 - The gazetteer you used covers a good and broad selection of terms. Was there a specific reason why some terms eg Xmen were included and not others that I imagine would be prevalent such as Marvel? Similarly, were some terms used found to not be as relevant as may have been expected from existing documentations you used to develop your gazetteer, eg blackberry vs iPad, in which case the billionse documents and guidelines may require an update?	This limitation is acknowledged and expanded on in the discussion: "There are limitations to the rule-based approach with such a rapidly evolving field. We endeavoured to have a broad range of search terms in the gazetteer, but acknowledge that this is not exhaustive. New games, social media platforms, websites and apps are hard to keep up with and omission of these titles from the gazetteer may bias studies towards certain online activities."
6 - I find the figure of 250,000 million documents (amounting to 250 billion) in line 35 page 5 hard to believe - could you confirm if this is accurate or should instead be 250 million documents?	This typo has been amended.

7 - Grammar: please could you check appropriate use of apostrophes eg "to the authors knowledge" (line 60 page 2) should be "to the authors' knowledge" (repeated several times in the manuscript)	I have changed the instances of this in the manuscript.
--	---

VERSION 2 – REVIEW

REVIEWER	Levis, Maxwell White River Junction VA Medical Center
REVIEW RETURNED	17-Apr-2023

GENERAL COMMENTS	Thanks for addressing our concerns. The manuscript is now much stronger and more robust.
--